# Mechanical Properties of AA2024 Aluminum/MWCNTs Nanocomposites Produced Using Different Powder Metallurgy Methods

**Fani Stergioudi** [1,2]**, Alexandros Prospathopoulos** [1,2]**, Alexandros Farazas** [1]**, Evangelos Ch. Tsirogiannis** [3] **and Nikolaos Michailidis** [1,2]*****

1 Physical Metallurgy Laboratory (PML), Mechanical Engineering Department, Aristotle University of Thessaloniki, 54124 Thessaloniki, Greece

2 Center for Research & Development of Advanced Materials (CERDAM), Center for Interdisciplinary Research and Innovation, Balkan Center, Building B', 10th km Thessaloniki-Thermi Road, 57001 Thessaloniki, Greece

3 EODH SA, 31 Giannitson str., Balkan Center, Filippos Building, 54627 Thessaloniki, Greece

\* Correspondence: nmichail@auth.gr; Tel.: +30-2310-995-891

**Abstract:** Metal matrix composites are a class of materials with high potential for industrial application due to the multifaceted properties that they possess. In the present research, mechanical alloying via ball milling was used to produce AA2024 powder that was subsequently reinforced with multiwalled carbon nanotubes (MWCNTs). Dispersion of the MWCNTs in the AA2024 matrix was achieved also by ball milling. Two different powder metallurgy sequences, (i) double pressing double sintering and (ii) hot pressing were used for compaction and consolidation of the AA2024 reinforced by MWCNTs. The produced nanocomposites outperform the pristine AA2024 in terms of compressive strength, elongation to failure, and microhardness. The strengthening mechanism was associated with the homogeneous distribution of MWCNTs in the AA2024 matrix and their efficient interfacial bonding, which was attested also by microstructural characterization. Hot pressing resulted in higher mechanical properties of the nanocomposite material. However, when reinforcement content was above 2 wt.% a dramatic decrease of mechanical properties was observed, attributed to clustering and inhomogeneous dispersion of the MWCNTs. The homogeneous dispersion of MWCNTs in the AA2024 matrix and the retaining of their structural integrity are pivotal in increasing the mechanical properties, which can be directly associated with the efficient interfacial load transfer between MWCNTs and AA2024 matrix.

**Keywords:** aluminum alloy; multiwall carbon nanotubes (MWCNTs); mechanical properties; microstructure

## 1. Introduction

In the past decade, carbon nanotubes (CNTs) have attracted a lot of scientific and industrial attention due to their inherent advanced mechanical properties combined with being extremely light weight. The use of CNTs as reinforcement material for the production of metal matrix composites (MMCs) is anticipated to be very promising, especially in potential applications in transportation and defense industry (for example use in armored combat vehicles), where the combination of lightweight materials with enhanced mechanical properties is required. The singly dispersed CNTs formed compact bonding with the matrix, which contributed to the grain refinement and the mechanical properties enhancement. The strengthening contributions are based on the grain refinement, load transfer, and Orowan looping mechanisms [1]. Although single-walled CNTs have been reported to possess extremely high elastic modulus (as high as 1.8 TPa) along with enhanced

mechanical properties (from 50 GPa to as high as 150 GPa), usually MWCNTs are mostly examined as reinforcement materials because of the higher efficiency and ease of production over the single-walled ones [2,3].

As summarized in the recent review of Khanna et al. [2], numerous works are reported in the literature that focus on the reinforcement of aluminum or aluminum alloys through powder technology. Nevertheless, several difficulties related to the inherent properties of CNTs, continue to impede the growth and development of efficient CNT-Al alloys composites with advanced properties. The tendency to cluster, the poor wettability of CNTs with Al, and the formation of a variety of different phases such as aluminum carbide ($Al_4C_3$) or aluminum oxides at temperatures over 450 °C are the main problems that need to be addressed [2–5]. Overall, the mechanical properties of CNTs-reinforced aluminum alloys are associated with uniform dispersion of the CNTs, the aspect ratio and quality of CNTs used, and the interfacial bonding between Al and CNTs [2,6].

Efforts have been directed to improve the dispersion of CNTs in the metallic matrix either by employing several chemical pretreatments on the CNTs [5–12] and using co-additions in the CNTs-powder mixture (such as nanoparticles or graphite) [13,14] or by improving the shape and surface characteristics of the initial metal powders used [8]. Ball milling is commonly applied; however, the long duration of the milling process imposes severe damage to CNTs which typically leads to inferior mechanical properties of the CNT/Al composite [2,10,11,15–17]. For the compaction and consolidation of powder mixtures cold pressing [15,18], hot pressing [13,19], hot extrusion [6,8,11,20,21], spark plasma sintering [7,10] are typically used to produce the composite Al alloys reinforced with CNTs.

In the present work, Al2024 was successfully reinforced by MWCNTs by utilizing a production method that combines mechanical alloying, milling, and two different compaction and sintering routes. The proposed production methods use industrial manufacturing techniques that can potentially lead to large-scale manufacturing of added-value AA2024-MWCNTs nanocomposites for use in demanding lightweight structures, such as in armored combat vehicles. The goal was to achieve a uniform dispersion of the MWCNTs in the metal matrix and simultaneously retaining minimum structural damage of the MWCTs by using powder technology techniques that can potentially lead to commercial mass production.

## 2. Materials and Methods

MWCNTs having purity higher than 95% and diameter between 50 and 90 nm were used in this study. The aspect ratio of length to diameter was higher than 100. The alloy of AA2024 was produced via mechanical alloying by using elemental powders at a nominal composition of 4 wt.% Cu, 1.3 wt.% Mg, 0.5 wt.% Mn, Al-balance. All the above materials were purchased from Sigma/Aldrich company (St. Louis, MO, USA).

The production of AA2024 mix, used as the matrix material, was carried out via a high energy vertically stirred ball mill at a rotation speed of 500 rpm for 3 h in a continuously flowing argon atmosphere to prevent oxidation. Stearic acid (1 wt.%) was added to the mixture as a process control agent while few drops of ethanol were also added to prevent from powder agglomeration. During the mechanical alloying process, stainless-steel balls with a ball to powder weight ratio of 10:1 were employed.

The production of the nanocomposite material was also achieved by ball milling. Different weight fractions of MWCNTs (1, 2, 3, 4, 5, and 6 wt.%) were subjected to a mixing process with the previously milled AA2024 powder by using the same stirred ball mill and similar milling conditions. However, the rotation speed was reduced to 250 rpm whilst the milling duration was kept to 60 min to prevent from damage of the MWCNTs.

The milled powder mixtures consisting of AA2024 and the MWCNTs were uniaxially compacted in a cylindrical stainless-steel mold having an inner diameter of 18 mm. A graphite suspension was employed as a lubricant to reduce friction effects. Two different

production methods were employed during compaction, (i) a double pressing-double sintering (DPDS) method and (ii) a hot pressing (HP) method.

For the DPDS method, the milled composite powders mixtures were first hot compacted under a pressure of 250 MPa at a constant temperature of 400 °C for 10 min and then pre-sintered at a low vacuum furnace (0.1 mbar) at 520 °C for 60 min employing a Dentsply NEYTech Centurion VPC (Charlotte, NC, USA). Before pressing of green compacts for the second time, they were minimally sprayed with a graphite suspension to reduce friction with the mold. The second hot compaction was performed under a pressure of 550 MPa at 400 °C for 10 min. Subsequently, the green compacts were sintered under low vacuum atmosphere at 590 °C for 120 min.

For the HP method, the milled composite powders mixtures were hot pressed at 400 MPa under 550 °C for 60 min. Then, the green compacts were sintered under low-vacuum atmosphere at 590 °C for 120 min. For reasons of comparison, the milled AA2024 was also hot pressed and sintered under the same conditions. This sample was designated as pristine AA2024.

X-ray diffraction (XRD) was used to determine the crystal structure of milled composite powders and sintered composite samples. Differential scanning calorimeter (DSC) using a TA Instruments DSC 25 (New Castle, DE, USA) was applied to determine phase transformation during sintering process from room temperature to 600 °C at a heating rate of 15 °C/min. Scanning electron microscopy (SEM) employing a Thermo Fisher Scientific Phenom ProX (Waltham, MA, USA) under an acceleration voltage of 10 kV was employed for microstructural characterization evaluation and for determining the distribution of different phases in the sintered composite samples.

The density of the produced nanocomposites was estimated using the Archimedes method and the equation (1)

$$\rho_c = (m/(m - m_1) \times \rho_w \tag{1}$$

where $m_1$ is the mass of the produced nanocomposite in distilled water, m the mass of the composite in air, and $\rho_w$ the density of the distilled water equal to 1 g/cm³.

The theoretical density of the produced nanocomposites was estimated using the rule of mixtures and the equation (2).

$$\rho_c = \rho_{CNTs} \times V_{CNTs} + \rho_{AA} \times V_{AA} \tag{2}$$

where $\rho_{CNTs}$ is the true density of the MWCNTs equal to 1.7 g/cm³, $V_{CNTs}$ the volume fraction of the MWCNTs in the nanocomposite, $\rho_{AA}$ the density of AA2024 equal to 2.78 g/cm³ and $V_{AA}$ the volume fraction of the AA2024 in the nanocomposite.

The mechanical response of the sintered composite specimens was characterized using a Shimadzu type M (Schimadzu, Kyoto, Japan) Vickers micro-hardness tester under a load of 100 g and dwell time of 15 s. The micro-hardness values are obtained as the average of at least five measurements. Quasi- static compression tests were also performed at a strain rate of 0.01 s⁻¹, which is a typical value for low modulus materials. The compressive tests were repeated in three samples per condition tested to attain reproducible and reliable results. The compressive stress–strain properties were obtained as an average of three measurements. The dimensions for the solid compression test specimens were selected from the suggested ones in table 2 of ASTM E9 (Standard Test Methods of Compression Testing of Metallic Materials at Room Temperature). According to ASTM E9, short specimens (length to diameter ratio in the range between 0.8 to 2) are typically used for compression tests of such materials as bearing metals, which in service are used in the form of thin plates to carry load perpendicular to the surface. The length of all cylindrical samples used for compression tests was 20 mm and their diameter 16 mm (L/D=1.25). Fractured surfaces were also examined by SEM, using fragments of specimens that were extracted from the compressed samples.

## 3. Results and Discussion

### 3.1. Structural Characterization of the Milled Composite and Sintered Composite

The initial mix of powders comprising the AA2024 alloy applied in this study are shown in Figure 1a. The obtained AA2024 powder after the ball-milling process presented a flake-shaped morphology (Figure 1b,c), which has been reported to be favorable for the incorporation of MWCNTs to the matrix [8,22]. The MWCNTs used are depicted in Figure 1d,e. An indicative SEM image of the ball-milled powder mixtures consisting of AA2024 and the MWCNTs is shown in Figure 1f. The MWCNTs were uniformly dispersed and incorporated into the AA2024 powders as no aggregates of them were visible. Although the MWCNTs appear to be fragmented and smaller (arrows in figure 1f) than the original ones used, most of them retain an adequate length, which is estimated between 8 and 15 µm, indicating that the damage of the MWCNTs during ball milling was not severe.

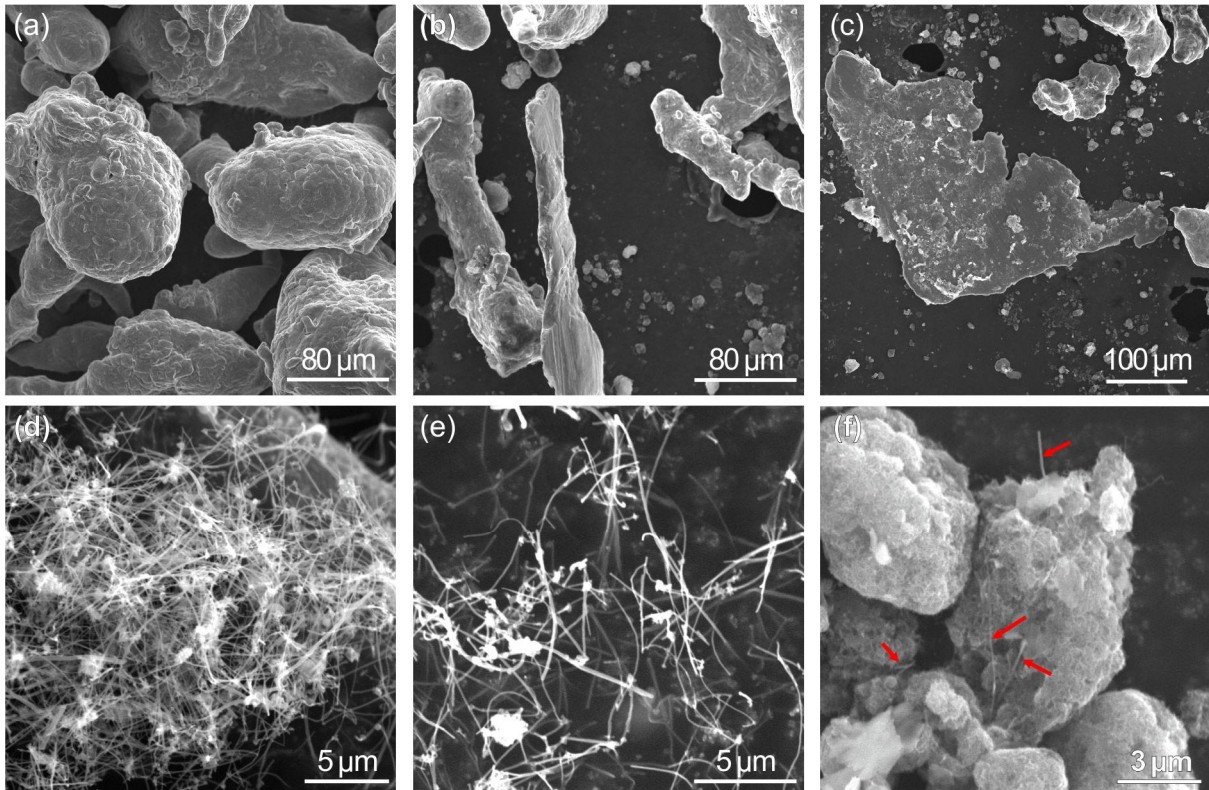

**Figure 1.** SEM images of (**a**) initial AA2024 powders used, (**b**) and (**c**) AA2024 powders after ball milling, (**d**) and (**e**) MWCNTs used and (**f**) powder mixture of AA2024 and MWCNTs after ball milling.

The XRD patterns of the milled AA2024, the milled AA2024-MWCNTs nanocomposite and the MWCNTs used are presented in Figure 2. Concerning the XRD of MWCNTs, the characteristic peak at $2\theta = 26°$ is clearly visible which corresponds to C(002) graphite peak [15,19,23,24]. The high sharpness of this peak suggests almost perfect crystallinity of the 2D graphene layer that gives the tubular structure of the nanotubes. Additionally, the C(002) peak is not sifted toward higher values of $2\theta$ (higher than 26°) which is an indication of high purity of the MWCNTs. Some peaks of significant lower intensity also exist at $2\theta = 53°$ (denoted as the (004) peak of graphite) and $2\theta = 78°$ (denoted as the (hko) peaks of graphite) [23,24]. These peaks are not sharp which stipulate that the crystallinity of the several nanotubes consisting of the MWCNTs may vary. The broadening of this peak may also be an indication for the presence of other forms of graphitized carbon [15,19,23,24]. The peak at around $2\theta = 43°$ is ascribed to the catalyst (most probably to cobalt) and the coexistence of some other impure forms of carbon.

The diffraction peaks of aluminum are clearly visible in both milled AA2024 samples. Some peaks corresponding to copper and magnesium are also visible. When the AA2024 is milled with the MWCNTS for 2 h, the magnesium peaks disappear whilst only copper peaks are still visible. However, these peaks are less intense, and a broadening of their width is observed. This result suggests that the production of AA2024 was successfully accomplished via mechanical alloying. Concerning the MWCNTs, only the C(002) can be detected. The absence of the remaining C peaks indicated that the MWCNTS were obviously refined and some of the carbon content was dissolved into aluminum to form a solid solution. To verify this assumption, differential scanning calorimetry for the milled AA2024-2 wt.% MWCNTs was performed.

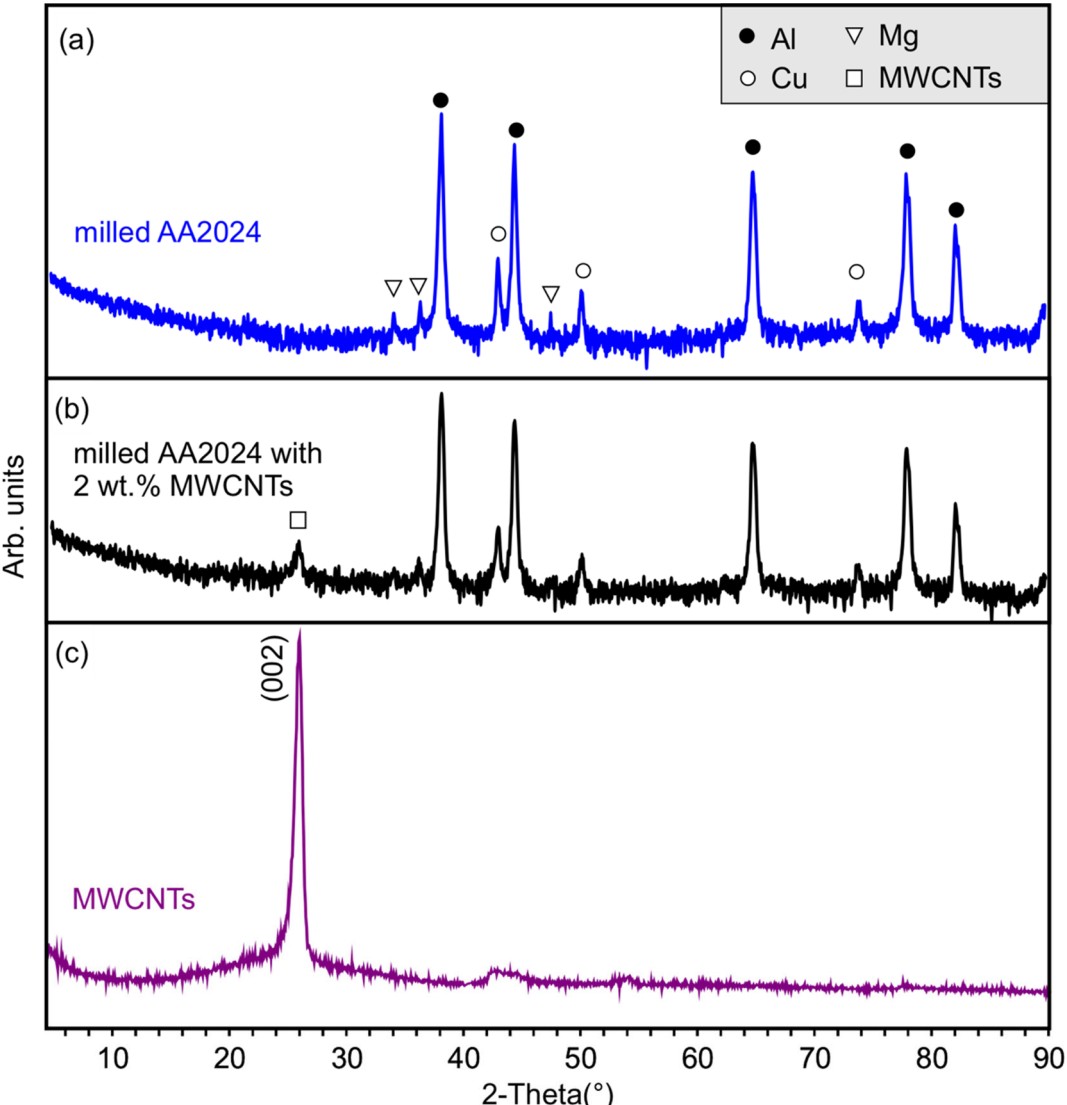

**Figure 2.** XRD patterns of (**a**) the milled AA2024, (**b**) the milled nanocomposite with 2% MWCNTs powder mixtures, and (**c**) the initial MWCNTs used.

The DSC curve is shown in Figure 3. The initial peak observed at low temperatures is attributed to humidity release and the evaporation of ethanol. The milled powder mixture of AA2024-MWCNTs appears to have a small exothermic peak in the range of 400 to 430 °C. This peak can be associated with the formation of $Al_4C_3$ [2,7,8]. However, the released amount of heat is quite small. Therefore, the formation of $Al_4C_3$ during sintering can be attributed to the decomposition of the metastable Al–C solid solution. Nevertheless, no visible broadening of the aluminum peaks was observed for the milled AA2024-

MWCNTs with respect to the milled AA2024 in the XRD patterns (Figure 2). This is an indication that aluminum has not been absorbed in its structure considerable amount of carbon content. The lattice parameter and the crystallite size remained stable for the Al in all cases examined, which confirmed that carbon content in the aluminum is negligible. Moreover, in the XRD patterns of the sintered samples no obvious Al₄C₃ peaks were identified in the milled and sintered AA2024-MWCNTs samples (Figure 4). This is evidence that the MWCNTs used in this study are stable in contact with Al and do not form aluminum carbides through a decomposition mechanism of the CNTs [25]. This is also an indication (along with the findings through XRD) that the MWCNTs used are of high quality and exhibit low content of defective nanotubes or carbon impurities [26].

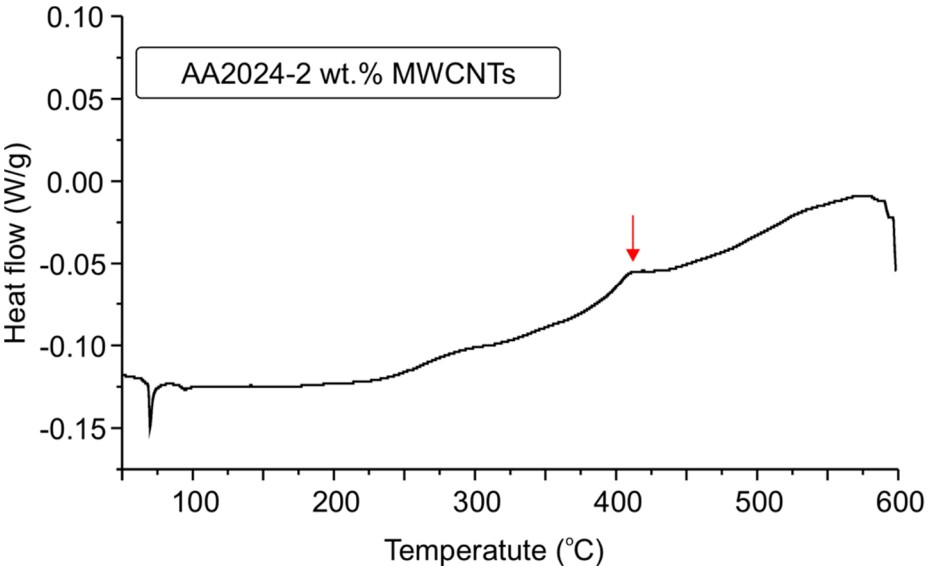

**Figure 3.** DSC curve for AA2024-2 wt.% MWCNT composite sample.

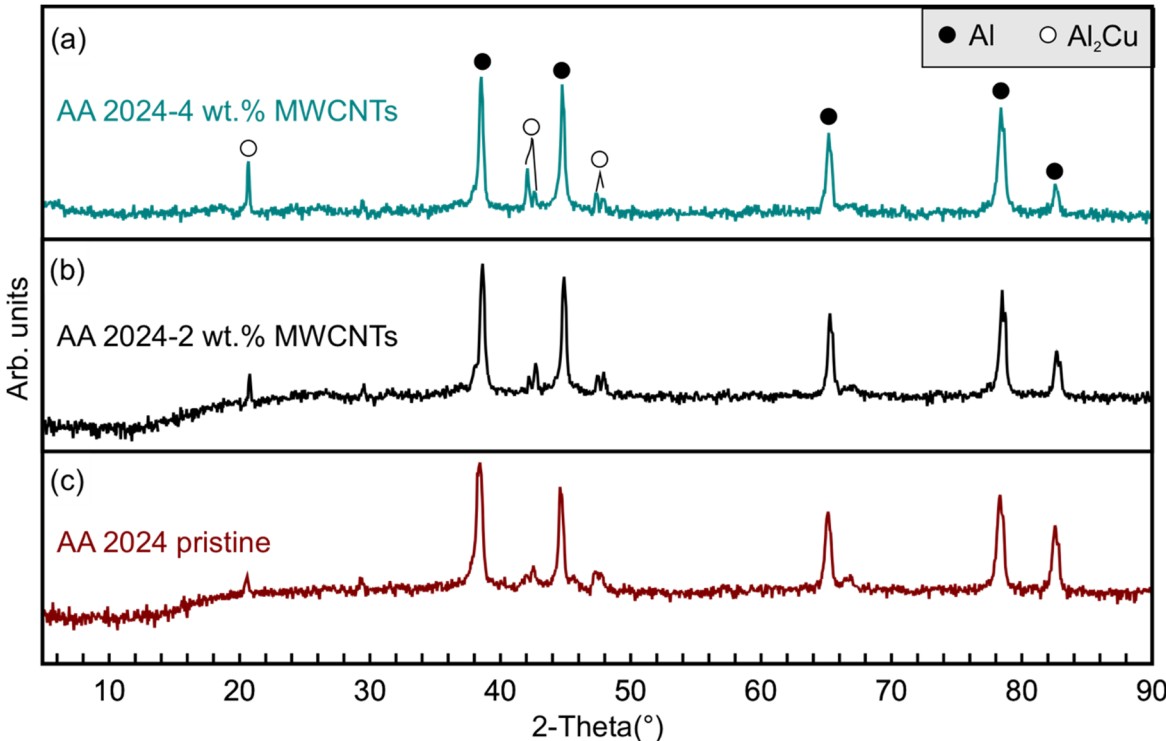

**Figure 4.** XRD patterns of milled and sintered (**a**) AA2024-4 wt.% MWCNTs and (**b**) AA2024-2 wt.% MWCNTs composites (HP method), as well as the (**c**) pristine AA2024.

Figure 4 shows the XRD patterns of milled and sintered AA2024-MWCNTs samples having two different MWCNTs compositions (a) 4 wt.% and (b) 2 wt.% respectively, produced via the HP method. Pristine sintered AA2024 is also shown (Figure 4c). After sintering, the peak of copper disappeared but an intermetallic phase $Al_2Cu$ is apparent at all sintered samples (both AA2024 and the composite) irrespectively of the MWCNTs content. However, these peaks present noticeable variations in their intensity as the MWCNTs content increases suggesting that the nanotubes play a significant role in the microstructural evolution of intermetallic phases in the AA2024 matrix.

In both XRD patterns (Figure 4a,b) of the composite materials, no C(002) peak of MWCNTs could be identified. This can be attributed to the small amount of existing MWCNTs on the surface sample examined and the small scattering length of carbon compared to metal atoms.

The microstructures of the produced AA2024-2 wt.% MWCNTs composite using both compaction routes are shown in Figure 5. Both methods provide satisfactory consolidation between the metal powders (Figure 5a,b). In both methods the presence of intermetallic $Al_2Cu$ phase is visible (Figure 5c,d), both in grain boundaries and at the inner regions of the grains as characteristically indicated with the lines. However, it seems that the HP methods favor the development of $Al_2Cu$ intermetallic phases which are highly visible and homogenously dispersed in the microstructure of the nanocomposite produced (Figure 5b). The flake morphology of the initial AA2024 powder is retained during the DPSD compaction method. HP results in a grain refinement (Figure 5d). The grains in that case are more equiaxed (with respect to the DPDS method). Additionally, subgrains are visible in the nanocomposite produced by the HP methods. As an example, some subgrains are marked with dashed red lines in Figure 5d. This observation led to the conclusion that the HP method resulted in grain refinement by two different mechanisms, i) flattening of the flaked morphology of the ball milled powders used, which led to grain size reduction of the initial grains produced by the AA2024 powders and ii) creation of subgrain inside these grains indicating grain refinement. This phenomenon is ascribed to dynamic recovery and recrystallization driven by the high plastic strain and heat during HP [27,28]

On the contrary, for the DPDS method only the initial grains produced by the consolidation of the AA2024 powders are obvious. A higher magnification of the black regions that appear in Figure 5c and 5d is shown in Figure 5e and 5f, respectively. It seems that these regions correspond to Al oxides since they are rich in oxygen (as indicated by EDS analysis). These oxides were probably created during the hot compaction stage of both production methods since this stage was performed in air. Any porosity in this oxide is probably caused during the metallographic preparation of the sample, resulting in the fracture of such brittle phases. The fact that the oxides have different shape is attributed to the argon atmosphere used in the DPDS method. It is reported in [29] that in Al-CNTS composites when a neutral atmosphere is used during HP, the amorphous $Al_2O_3$ is transformed into coarse lump shaped crystalline $\gamma$-$Al_2O_3$ at matrix grain boundaries. The presence of the oxide in the AA2024 grains is pivotal, as it prevents the MWCNTs to react with Al and produce brittle aluminum carbides {29}.

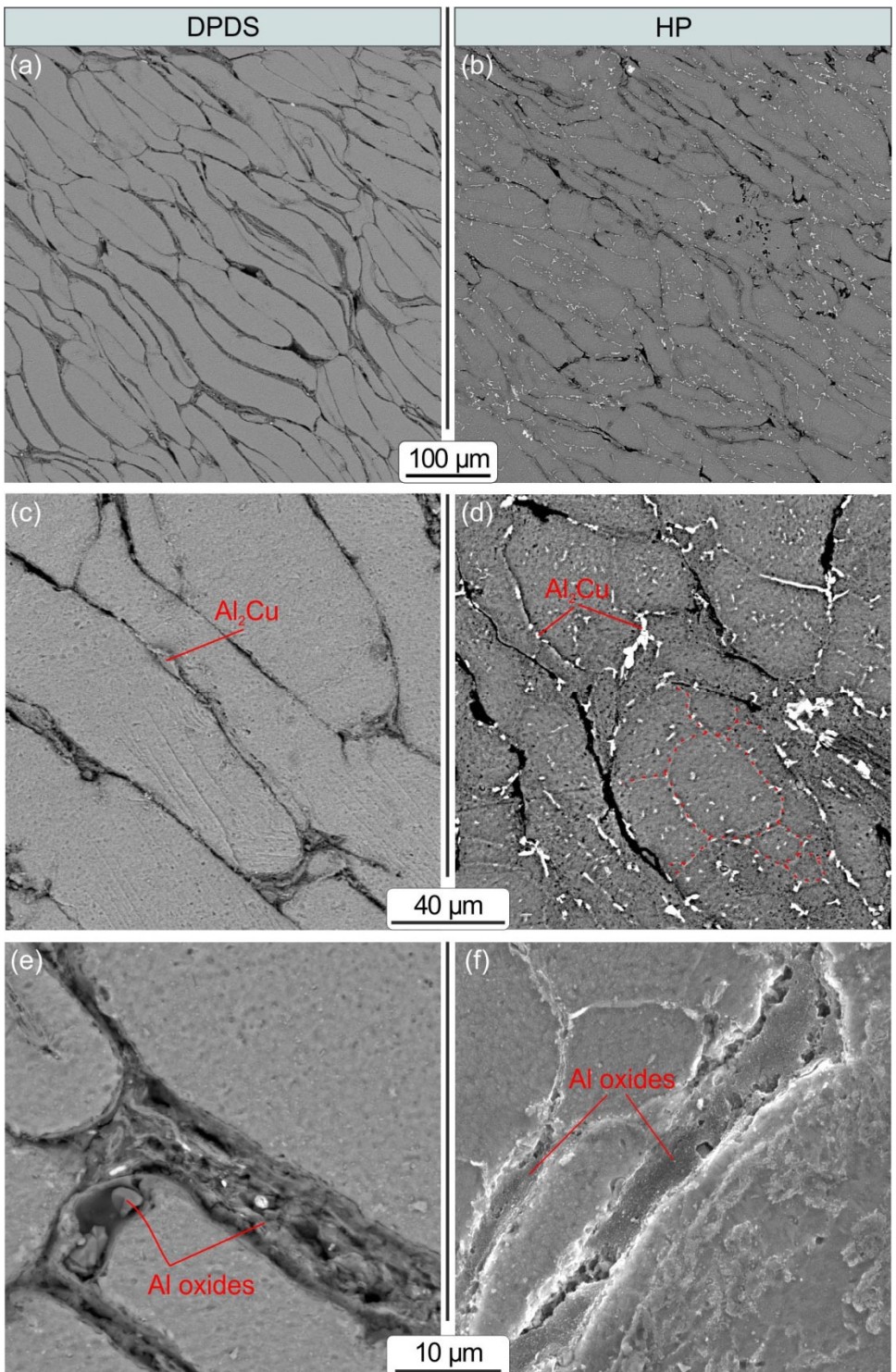

**Figure 5.** Microstructure of the nanocomposite AA2024-2 wt.% MWCNTs produced via DPDS (**a**), (**c**) and (**e**) and HP, (**b**), (**d**), and (**f**).

### 3.2. Mechanical Characterization

The compressive stress–strain curves of the AA2024-MWCNTs nanocomposites produced with DPDS method at various MWCNTs contents are presented in Figure 6a. The addition up to 2 wt. % MWCNTs results in enhancement of mechanical properties of the nanocomposite. However, higher MWCNTs contents than 2 wt.% lead to a dramatic decrease of the mechanical properties, as shown in Figure 6a,b for the case of AA2024-4% MWCNTs nanocomposite. This behavior can be attributed to an inhomogeneous

distribution of the nanotubes in the AA2024 matrix, and their clustering, thus leading to low interfacial bonding between the AA2024 and the MWCNTs.

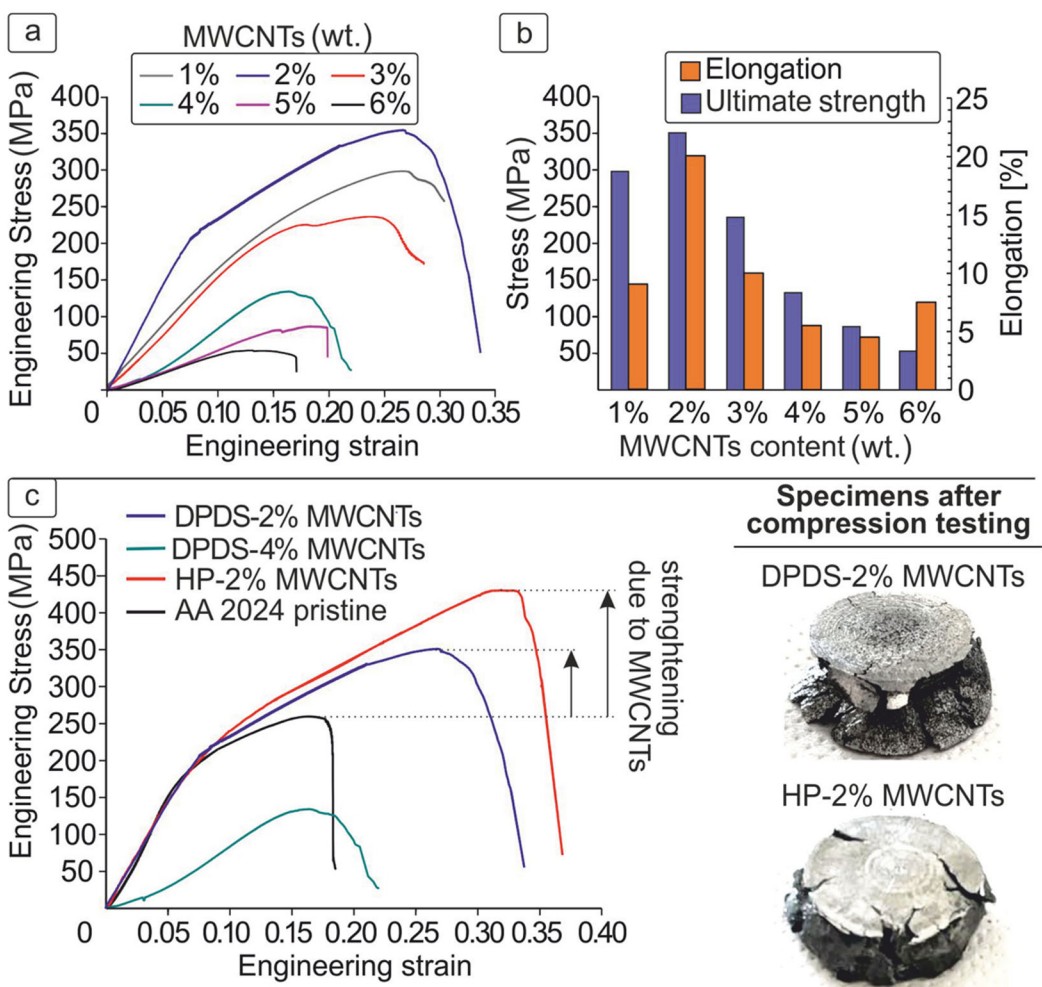

**Figure 6.** (**a**) Compressive stress–strain curves and (**b**) strength and elongation values for various AA2024 nanocomposites with addition of 1, 2, 3, 4, 5, 6 wt.% MWCNTs produced with DPDS method. (**c**) Compressive stress–strain curves of the pristine AA2024 and several AA2024-MWCNTs produced under different conditions along with indicative photos of specimens post compression.

Compressive stress–strain curves of the pristine AA2024 and various AA2024-MWCNTs nanocomposites produced under different conditions along with indicative photos of specimens post compression testing are shown in Figure 6b. Pristine AA2024 is also shown for reasons of comparison (Figure 6b). For the composite AA2024-2% MWCNTs, both production methods, DPDS and HP led to a significant increase in yield stress and ultimate strength compared to the pristine AA2024. The fracture elongation was also remarkably raised from 0.11 for the pristine AA2024 to 0.17 for the DPDS method and 0.19 for the HP method, respectively. This ductility increase is ascribed to the homogeneous dispersion of MWCNTs and the strong interfacial bonding between MWCNTs and AA2024 matrix [2,4].

The HP compaction method leads to higher mechanical properties of the nanocomposite (with the exception of Young's modulus) compared to DPDS method. This is ascribed to grain refinement of the matrix material (see Figure 5f) and higher density of the composite produced. More specifically, the density of the composite produced by DPDS method and by HP method was 2.5 gr/cm³ (corresponding to 92.3% of the theoretical one) and 2.65 gr/cm³ (corresponding to 98% of the theoretical one), respectively. Regarding the strength-ductility efficiency (compressive strength X elongation) for the AA2024-2 wt.%

MWCNTs produced by DPDS, it was calculated to be 5.25 GPa and for the one produced by HP is 8.17 GPa, which indicates that for the case of HP the balance between high strength and ductility was successfully achieved [30].

The combined effects of the AA2024 matrix and MWCNTS reinforcement, namely the strengthening of grain boundaries, grain refinement, load transmission from matrix to MWCNTs, and Orowan looping are the possible mechanisms that result in the strengthening of the produced nanocomposites [27]. CNTs alone exhibit significant Orowan strengthening when the matrix grains are significantly bigger than the reinforcements, and the CNTs are mainly located on the grain boundaries, which is applicable in our case [27].

An efficient load transfer from the AA2024 matrix to the MWCNTs is an important condition to effectively utilize the superior mechanical properties of the nanotubes. Their interface with the matrix and consequently the mechanical properties are directly associated with the even dispersion of the nanotubes in the metal matrix, the structural integrity of MWCNTS, and interfacial bonding of CNTAl [4,10,27]. The interaction of dislocations and the nanotubes could also be a favorable mechanism leading to reinforcement of the composites [4,25]. In order to check the load transfer, a fractographic analysis of compressed samples was performed.

Micro-hardness values of the composite materials are consistent with the results of the mechanical compressive testing showing higher values for the AA2024-2 wt.% MWCNTs composites compared to pristine AA2024 (Table 1). The considerably lower mechanical behavior of the AA2024-4 wt.% MWCNTs nanocomposite is also attested by micro-hardness. During micro-hardness testing no cracking was observed in the samples, which is a further indication of efficient load transfer.

**Table 1.** Vickers micro-hardness values of pristine AA2024 and several AA2024-MWCNTs composites produced under different conditions.

| Materials | Pristine AA2024 | DPDS-2 wt.% MWCNTs | DPDS-4 wt.% MWCNTs | HP-2 wt.% MWCNTs |
|---|---|---|---|---|
| Microhardness $HV_{0.1}$ | 56 | 62 | 49 | 70 |

SEM images of the fractured surfaces for AA2024-MWCNTs nanocomposites with 2 wt.% MWCNT content and 4 wt.% MWCNTs are shown in Figure 7a and 7b, respectively. Long pull-outs of MWCNTs were visible in the fractured surface of AA2024-2 wt.% MWCNT (indicated arrows), with a pull-out length of about 3 μm, indicating efficient load transfer at the CNT/matrix interface (Figure 7a). It should be noted that pull-outs are generally visible during tensile testing and present indications about the load transfer at the MWCNT/matrix interface and about the deployment of CNTs during the sample fracture [26]. Therefore, it is deduced that although pure compressive loading was applied macroscopically, some areas were loaded to shearing and tension. According to Stein et al. long pull-outs denote strong bonding between the matrix and the CNTs thus resulting to larger strengthening effects [26]. Despite the milling procedure, the cylindrical integrity over the length of the MWCNTs was preserved as evidenced by the fractured surfaces of both composite samples. The fact that the nanotubes are not considerably shortened during the milling procedure is confirmed by these lengthy pull-outs. It is worth noting that the existence of pull-out MWCNTs is related to their distribution in the fractured surfaces of the samples under investigation.

Clustering of MWCNTs and inhomogeneous distribution in the matrix is visible in the fractured surface of AA2024-4 wt.% MWCNT nanocomposite (Figure 7b). This MWCNTs agglomeration leads to weak interface bonding between the MWCNTs and the AA2024 matrix and subsequently to inferior mechanical properties of the composite. Low content of CNTs reinforcement has been also reported by other researchers to yield a more

homogeneous dispersion of CNTs in the matrix, thus setting an upper limit in the content of the incorporated reinforcement [4,6,26,31].

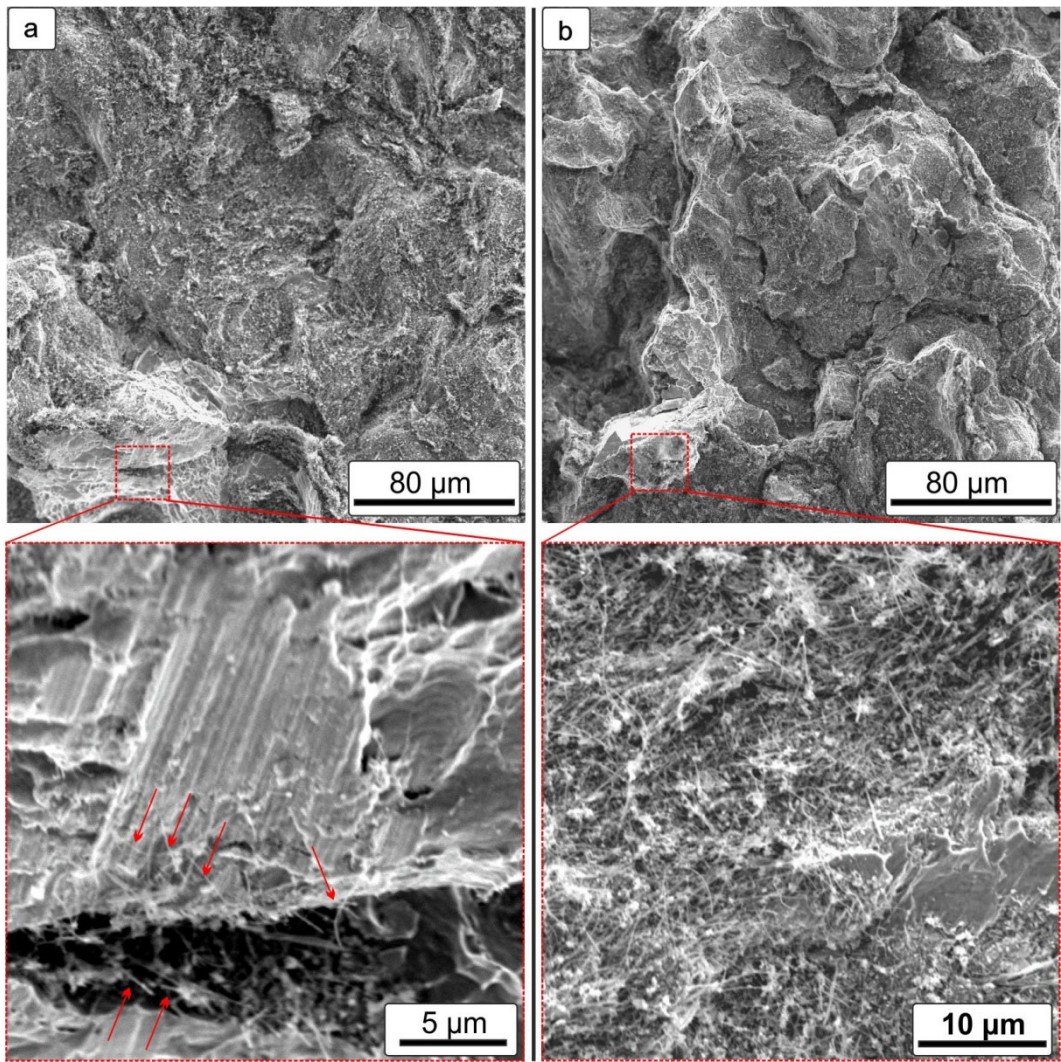

**Figure 7.** SEM images of fractured surfaces for (**a**) AA024 - 2 wt.% MWCNT and (**b**) AA2024-4 wt.% MWCNTs produced through DPDS method.

## 4. Conclusions

An AA2024 composite reinforced with MWCNTs has been successfully produced using a manufacturing approach that combines mechanical alloying and milling followed by two different compaction and sintering processes (DPDS and HP). The mechanical properties of the produced nanocomposites were investigated and correlated with microstructural morphologies. The following conclusions can be drawn:

1) Both compaction techniques used to produce the composite material resulted in reinforcement of the mechanical properties with respect to pristine AA2024. However, HP was proven to be superior which was mainly ascribed to the higher density of the obtained composite and to grain refinement. The structural integrity of the MWCNTs was retained, although some fracturing of the MWCNTS was observed which also contributed to efficient load transfer from the matrix to the MWCNTs.

2) A strong link between MWCNTs content and mechanical strength was established. A dramatic decrease of the mechanical properties is reported above 2 wt.% MWCNTs, which is attributed to clustering of MWCNTs.

3) The compressive strength of the AA2024-2 wt.% MWCNTs nanocomposite reached 350 MPa when produced by DPDS method and 423 MPa when produced by HP

method. The elongation reached the values of 16% and 19%, respectively, indicating that both methods can provide nanocomposite with satisfactory strength-ductility efficiency.

**Author Contributions:** Conceptualization, E.C.T.; methodology, F.S. and N.M.; formal analysis, F.S. and N.M.; investigation, A.P., A.F.; resources, N.M.; data curation, A.P., A.F.; writing—original draft preparation, F.S.; writing—review and editing, N.M., F.S.; visualization, F.S., A.P.; supervision, N.M., F.S.; project administration, N.M.; funding acquisition, E.C.T. All authors have read and agreed to the published version of the manuscript.

**Funding:** This research has been co-financed by the European Regional Development Fund of the European Union and Greek national funds through the Operational Program Competitiveness, Entrepreneurship and Innovation, under the call RESEARCH–CREATE–INNOVATE (project code: T1EDK-04420).

**Institutional Review Board Statement:** Not applicable

**Informed Consent Statement:** Not applicable

**Data Availability Statement:** Not applicable

**Conflicts of Interest:** The authors declare no conflict of interest.

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
