# Peer review of "Mechanical Properties of AA2024 Aluminum/MWCNTs Nanocomposites Produced Using Different Powder Metallurgy Methods"

_metals, doi:10.3390/met12081315_

Round 1

Reviewer 1 Report

In the work, the AA2024 composite reinforced with MWCNTs was produced by mechanical alloying and milling followed by two different compaction and sintering processes (DPDS and HP). The homogeneous dispersion of MWCNTs increased mechanical properties in the AA2024 matrix. The proposed methods can potentially lead to large-scale manufacturing of added-value AA2024-MWCNTs nanocomposites for use in demanding lightweight structures. The experimental design was innovative, and the results are solid. The manuscript was well written. The reviewer believes that this is a paper of high quality. It is suggested that the manuscript be accepted after the following minor comments are taken care of.

1.      Sound and concise conclusions and abstract with the supporting results should be given to the reader.

2.      The use of CNTs as reinforcement material for the production of metal matrix composites (MMCs) is anticipated to be very promising, especially in potential applications in transportation and defense industry, where the combination of lightweight materials with enhanced mechanical properties is required. The singly dispersed CNTs formed compact bonding with the matrix, which contributed to the grain refinement and the mechanical properties enhancement. The strengthening contributions are based on the grain refinement, load transfer and Orowan looping mechanisms[https://doi.org/10.1016/j.msea.2018.07.011].

3.      The authore claimed that the hot pressing results in a grain refinement. The grains in that case are more equiaxed. The direct supporting results with detail discussion should be added.

4.      Abbreviations should be given their full names when they appear for the first time. And their full names should not used in the following text repeatly.

5.      For the AA2024-MWCNTs nanocomposites, how can get the strength-ductility efficiency? Why? The in-depth discussion should be added. The deformation-driven metallurgy with high plastic strain and frictional/deformation heat contributes to dynamic recovery and recrystallization [https://doi.org/10.1016/j.compscitech.2021.109225]. And The stability of the ultra-fine-grained microstructure was improved under the effect of the Zener pinning mechanism for aluminum matrix composites via deformation-driven metallurgy[https://doi.org/10.1016/j.jallcom.2021.162078]. Via deformation-driven metallurgy under coupled thermo-mechanical effect, the severe plastic deformation and corresponding frictional/deformation heat contributed to solid-state and microstructural modification has inhibited the growth of recrystallized grains effectively with fine-grain microstructures to realize high strengthening-toughening efficiency [https://doi.org/10.1016/j.coco.2021.100776]. The strengthening-efficiency mechanisms are suggested to consider. The higher MWCNTs contents than 2 wt.% lead to a dramatic decrease of the 202 mechanical properties. Why?

Author Response

We would like to thank the reviewer for the constructive remarks that helped improve the quality of the manuscript. Appropriate changes and corrections were made in the manuscript to address all the proposed remarks. All changes are marked with yellow in the revised version of the manuscript.

Point 1: Sound and concise conclusions and abstract with the supporting results should be given to the reader.

Response 1: The conclusion section and the abstract were re-written as proposed.

Point 2: The use of CNTs as reinforcement material for the production of metal matrix composites (MMCs) is anticipated to be very promising, especially in potential applications in the transportation and defense industry, where the combination of lightweight materials with enhanced mechanical properties is required. The singly dispersed CNTs formed compact bonding with the matrix, which contributed to the grain refinement and mechanical properties enhancement. The strengthening contributions are based on the grain refinement, load transfer, and Orowan looping mechanisms[https://doi.org/10.1016/j.msea.2018.07.011].

Response 2: The proposed change was adapted to the manuscript (see the introduction section).

Point 3: The authors claimed that the hot pressing results in grain refinement. The grains in that case are more equiaxed. The direct supporting results with detailed discussion should be added.

Response 3: Appropriate discussion was added to the manuscript. Figure 5 was changed to support the results. New SEM images were added for both production methods.

Point 4: Abbreviations should be given their full names when they appear for the first time. And their full names should not used in the following text repeatly.

Response 4: Appropriate alterations were made throughout the manuscript regarding the abbreviations, as proposed.

Point 5: For the AA2024-MWCNTs nanocomposites, how can get the strength-ductility efficiency? Why? The in-depth discussion should be added. The deformation-driven metallurgy with high plastic strain and frictional/deformation heat contributes to dynamic recovery and recrystallization [https://doi.org/10.1016/j.compscitech.2021.109225]. And The stability of the ultra-fine-grained microstructure was improved under the effect of the Zener pinning mechanism for aluminum matrix composites via deformation-driven metallurgy[https://doi.org/10.1016/j.jallcom.2021.162078]. Via deformation-driven metallurgy under coupled thermo-mechanical effect, the severe plastic deformation and corresponding frictional/deformation heat contributed to solid-state and microstructural modification has inhibited the growth of recrystallized grains effectively with fine-grain microstructures to realize high strengthening-toughening efficiency [https://doi.org/10.1016/j.coco.2021.100776]. The strengthening-efficiency mechanisms are suggested to consider. The higher MWCNTs contents than 2 wt.% lead to a dramatic decrease of the 202 mechanical properties. Why?

Response 5: The strength-ductility efficiency was added to the manuscript. An appropriate discussion along with referencing was also added (see section 3.2 Mechanical characterization). The higher content of MWCNTs was attributed to the inhomogeneous distribution of the nanotubes in the AA2024 matrix and to their agglomeration.

Reviewer 2 Report

In this manuscript, the authors produced a metal matrix composite of AA2024 alloy reinforced by multi-walled carbon nanotubes. Two different powder metallurgy sequences were used, and the produced composites were compared in terms of mechanical properties and deformation mechanisms. This work is helpful for the industrial production. However, some critical issues need to be addressed in order to enhance the quality of this work. The reviewer's comments and suggestions are listed below:

1.     Authors mentioned at line 82, page 2, “the obtained AA2024 powder had a flake-shaped morphology after MA.” If the morphologies of the powders after ball milling, including AA2024 and MWCNTs, can be provided, the details of the preparation process will be more explicit.

2.     At line 85, page 2, “different weight fractions of MWCNTs (1,2,3,4,5 and 6 wt.%) were subjected to a mixing process…”. However, the article only presents the results of the composites with 2% and 4% weight fractions, and there is no description for the composites with other weight fractions. Are there any experimental results for these composites (1,3,5,6 wt.%)? Do they help to draw the conclusions of the article?

3.     In the materials and methods section, the description of time is mixed with statements such as 1 hour, 60 min, and 120 min. Please consider unifying to the same unit.

4.     For the quasi-static compression, a strain rate of 0.01 s-1 is relatively high, and the aspect ratio of the specimen is lower than 2. Why these experimental parameters were chosen and whether they affect mechanical properties. More explanations are needed here.

5.     There are many spelling mistakes in the manuscript. Such as page 3, line 125 “sifted” (shifted) and line 127 “hko” (hkl). Please check them carefully.

6.     Authors claimed at line 199, page 6, “This ductility increase is ascribed to the superior elongation of MWCNTs and the strong interfacial bonding between MWCNTs and AA2024 matrix.” The discussions on the interfacial bonding between WMCNTs and matrix on the mechanical properties should be improved, and more details need to be added.

7.     Although some SEM images have been presented, the characterization of the interface is still inexplicit. Therefore, if possible, please further examine the interfacial bonding.

8.     At line 218, Page 7, the authors claimed that the HP compaction method leads to higher mechanical properties ascribed to grain refinement of the matrix. But from the SEM images in Figure 4, the grain size of the HP method appears to be larger than that of the DPDS method. Please give further explanation.

Author Response

We would like to thank the reviewer for the constructive remarks that helped improve the quality of the manuscript. Appropriate changes and corrections were mede in the manuscript to address all the proposed remarks. All changes are marked with yellow in the revised version of the manuscript.

Point 1: Authors mentioned at line 82, page 2, “the obtained AA2024 powder had a flake-shaped morphology after MA.” If the morphologies of the powders after ball milling, including AA2024 and MWCNTs, can be provided, the details of the preparation process will be more explicit.

Response 1: The morphologies of the initial powders used, the powders after ball-milling, as well as after mixing with MWCNTs were provided in figure 1. The MWCNTs used were also provided in the same figure.

Point 2: At line 85, page 2, “different weight fractions of MWCNTs (1,2,3,4,5 and 6 wt.%) were subjected to a mixing process…”. However, the article only presents the results of the composites with 2% and 4% weight fractions, and there is no description for the composites with other weight fractions. Are there any experimental results for these composites (1,3,5,6 wt.%)? Do they help to draw the conclusions of the article?

Response 2:. Stress-strain curves for samples having 1, 2, 3, 5, 6 wt.% MWCNTs were presented in figure 6. It was explained that the 2 wt. % MWCNTs content results in optimum strengthening of the nanocomposite therefore the comparison of the two production methods was done only for the nanocomposites having 2 wt. % MWCNTs

Point 3: Ιn the materials and methods section, the description of time is mixed with statements such as 1 hour, 60 min, and 120 min. Please consider unifying to the same unit.

Response 3: Appropriate corrections were done.

Point 4: For the quasi-static compression, a strain rate of 0.01 s-1 is relatively high, and the aspect ratio of the specimen is lower than 2. Why these experimental parameters were chosen and whether they affect mechanical properties. More explanations are needed here

Response 4: Responses to the issues raised are discussed in the manuscript (lines 130-144).

Point 5: There are many spelling mistakes in the manuscript. Such as page 3, line 125 “sifted” (shifted) and line 127 “hko” (hkl). Please check them carefully

Response 5: The manuscript was thoroughly checked for spelling and typo mistakes. However, the hko is not a spelling error but an indication of certain peaks for the CNTs as stated in references 23 and 24, in order not to confuse them with the graphite peaks.

Point 6: Authors claimed at line 199, page 6, “This ductility increase is ascribed to the superior elongation of MWCNTs and the strong interfacial bonding between MWCNTs and AA2024 matrix.” The discussions on the interfacial bonding between WMCNTs and matrix on the mechanical properties should be improved, and more details need to be added

Response 6: The discussion in this section was improved. Appropriate referencing was also added.

Point 7: Although some SEM images have been presented, the characterization of the interface is still inexplicit. Therefore, if possible, please further examine the interfacial bonding

Response 7: Overall images of failed samples were added to figure 6. However, it was very difficult to find a better quality of SEM images of high magnification of the fractured surfaced due to blurring. Therefore, the authors decided not to alter these SEM images but to better explain them in the manuscript. We sincerely hope that the reviewer will understand that the task of finding CNTs in fractured surfaces of compressed samples and simultaneously obtaining a satisfactory SEM image of them is a demanding task of high complexity.

Point 8: At line 218, Page 7, the authors claimed that the HP compaction method leads to higher mechanical properties ascribed to grain refinement of the matrix. But from the SEM images in Figure 4, the grain size of the HP method appears to be larger than that of the DPDS method. Please give further explanation

Response 8: The images of figure 4 were replaced (current figure 5) to better visualize the grain refinement. The discussion was also added. Higher magnification images were also added showing that the black regions (initially translated as pores) correspond to Al-oxides stemming from the production methods. Any porosity in this oxide is probably caused during the metallographic preparation of the sample, resulting in the fracture of such brittle phases.

Reviewer 3 Report

The fabrication methods are explained well, and the topic is very interesting. However, the manuscript needs to be improved before publishing. Tensile or fatigue tests are more comprehensive testing for evaluating the mechanical properties of composite materials. The authors are encouraged to justify their choice of compression testing. The discussion of microhardness measurements and compression testing results should be presented according to the results, not speculation. For example, the explanation for low hardness measurements of 4% sample should be explained clearly, also, the details of testing, such as the number of the measurements and the standard deviation, are missing; the same information is required for the compression testing, e.g. the number of samples tested, etc.

The microstructural images presented in figure 4 should be replaced. The quality of the images presented is very poor. The images show some porosities; but no discussion was presented. Image analysis should be provided for area fraction of porosities and distribution of MWCNT. The fraction of porosities should be reported. How was the density measured?

The microscopy images of fractured surfaces using fragments of specimens that were detached from samples during compressive testing do not highlight the differences. Perhaps an overall image of the failed samples would be beneficial. The images do not clearly show the effect MWCNT.

The conclusions are just a summary of the results; it should be rewitten,

Author Response

We would like to thank the reviewer for the constructive remarks that helped improve the quality of the manuscript. Appropriate changes and corrections were made in the manuscript to address all the proposed remarks. All changes are marked with yellow in the revised version of the manuscript.

Point 1: Tensile or fatigue tests are more comprehensive testing for evaluating the mechanical properties of composite materials. The authors are encouraged to justify their choice of compression testing

Response 1: The choice for compression testing was justified in the manuscript. More specifically compression tests were chosen because the targeted application for the produced composites is in armored vehicles for the defense industry. In the authors' future works are included to evaluate the absorption capacity of the produced nanocomposite under dynamic compressive loading and compare it with the static behavior under static compressive loading.

Point 2: The discussion of microhardness measurements and compression testing results should be presented according to the results, not speculation. For example, the explanation for low hardness measurements of 4% sample should be explained clearly,

Response 2: Appropriate discussion was added to the manuscript (see section 3.2. Mechanical characterization)

Point 3: The details of testing, such as the number of the measurements and the standard deviation, are missing; the same information is required for the compression testing, e.g. the number of samples tested, etc.

Response 3: Details for compression testing such as the number of samples and number of measurements were added to the manuscript (last paragraph of the materials and methods section)

Point 4: The microstructural images presented in figure 4 should be replaced. The quality of the images presented is very poor. The images show some porosities, but no discussion was presented. Image analysis should be provided for the area fraction of porosities and distribution of MWCNT. The fraction of porosities should be reported. How was the density measured?

Response 4: The images of figure 4 were replaced as proposed (current figure 5). Discussion about porosity was added. More specifically higher magnification images were added showing that the black regions (initially translated as pores) correspond to Al-oxides stemming from the production methods. Any porosity in this oxide is probably caused during the metallographic preparation of the sample, resulting in the fracture of such brittle phases.

The method used to measure density was also added (last paragraph of the materials and methods section).

Point 5: The microscopy images of fractured surfaces using fragments of specimens that were detached from samples during compressive testing do not highlight the differences. Perhaps an overall image of the failed samples would be beneficial. The images do not clearly show the effect MWCNT.

Response 5: Overall images of failed samples were added to figure 6. However, it was very difficult to find better quality of SEM images with high magnification of the fractured surfaces due to blurring. Therefore, the authors decided not to alter these SEM images but to better explain them in the manuscript. We sincerely hope that the reviewer will understand that the task of finding CNTs in fractured surfaces of compressed samples and simultaneously obtaining a satisfactory SEM image of them is a demanding task of high complexity.

It was also explained in the manuscript that the piece used to make fractographic analysis was extracted from the compressed samples as a fragment.

For the reviewer, we present the figure below depicting the region from which the fragment was extracted.

Point 6: The conclusions are just a summary of the results; it should be rewitten

Response 6: The conclusion section was re-written as proposed.

Round 2

Reviewer 3 Report

I am happy with the response of the authors to the comments. I believe the work can be published in present form.